# A Low-Cost Synthetic Route of FAPbI_3_ Quantum Dots in Air at Atmospheric Pressure: The Role of Zinc Iodide Additives

**DOI:** 10.3390/nano13020226

**Published:** 2023-01-04

**Authors:** Shuo Wang, Simiao Li, Qian Zhao

**Affiliations:** School of Materials Science and Engineering, Nankai University, Tianjin 300350, China

**Keywords:** perovskite quantum dot, hot-injection synthesis, atmospheric pressure, ZnI_2_ additives, carrier lifetime

## Abstract

Perovskite quantum dots (PQDs) have shown great promise in optoelectronic device applications. Typically, a traditional hot-injection method with heating and high vacuum pressure is used to synthesize these colloidal nanoparticles. In this article, we report a low-cost synthetic method for FAPbI_3_ PQDs in air at atmospheric pressure with the assistance of ZnI_2_. Compared with the FAPbI_3_ PQDs synthesized under vacuum/N_2_ condition, the air-synthesized Zn:FAPbI_3_ PQDs exhibit the same crystalline structure with a similar preferential crystallographic orientation but demonstrate higher colloidal stability and higher production yield. Furthermore, we examine the influence of ZnI_2_ during the synthesis process on morphologies and optoelectronic properties. The results show that the mean size of the obtained FAPbI_3_ PQDs is decreased by increasing the amount of added ZnI_2_. More importantly, introducing an optimal amount of ZnI_2_ into the Pb source precursor enables increasing the carrier lifetime of FAPbI_3_ PQDs, showing the potential beneficial effect on device performance.

## 1. Introduction

Organic–inorganic hybrid perovskite with a general formula of ABX_3_ has attracted much attention over the past decade, in which A is a monovalent cation (such as methylammonium (MA^+^), formamidinium (FA^+^), or cesium (Cs^+^)), B is an inorganic metal cation (usually lead (Pb^2+^) or tin (Sn^2+^)), and X is a halide anion (chlorine (Cl^−^), bromine (Br^−^), or iodine (I^−^)) [1]. Due to its superior semiconducting characteristics, such as high carrier mobility, long carrier diffusion length, and high defect tolerance, metal halide perovskite has shown great potential as the active material in optoelectronic applications [2]. Among them, formamidinium lead triiodide (FAPbI_3_) has emerged as one of the most promising materials for highly efficient and stable perovskite solar cells, having achieved the record power conversion efficiency (PCE) of perovskite solar cells exceeding 25% [3,4,5,6,7,8,9,10]. Beyond photovoltaics, FAPbI_3_ has also shown a large potential in lighting and displays, owing to the high brightness and good color purity [11]. More recently, the quantum dot (QD) form of FAPbI_3_ has emerged as a promising material for next optoelectronic devices [12,13,14]. It additionally possesses many unique features from quantum dots such as wide tunable absorption range, multiple exciton generation, and high compatibility with a large-scale fabrication process [15].

Colloidal FAPbI_3_ perovskite quantum dots (PQDs) are generally synthesized by a room-temperature ligand-assisted reprecipitation process (LARP) or a hot-injection method [16,17]. In LARP, since the surface capping ligands are commonly hydrophobic, polar solvents such as DMF and DMSO must be used to flocculate the PQDs, which are subsequently isolated via centrifugation. However, the high toxicity of such polar coordination solvents makes LARP inapplicable in industrial applications, according to ecological requirements. Furthermore, the synthesized PQDs are vulnerable to polar solvent that cannot be used especially in photovoltaic devices. Therefore, the hot-injection method, without using polar solvent, has so far been considered to be the only synthetic route for PQD-based photovoltaics, also in terms of production yield and surface ligand manipulation particularly with short-chain ligands. During the hot-injection synthetic route, a vacuum degassing process (usually <20 Pa) is first applied to the FA-oleate solution and the Pb-source precursor for removing moisture and volatile contaminants, which is typically unavoidable [17,18]. Then, the degassed FA-oleate solution is rapidly injected into the hot Pb-source precursor with a nonpolar solvent under inert atmosphere. Lastly, the mixture is cooled in an ice bath [17,19]. As can be seen above, the degassing process and maintaining the inert atmosphere are not industrially feasible due to their high costs and large time consumption. Thus, seeking a facile route for low-cost, large-scale, and controlled synthesis at atmospheric pressure in air is essential when exploring FAPbI_3_ PQD-related practical applications.

Introducing active ligands or additives during synthesis has been proven to be effective for attaining PQDs with reduced defect density and improved structural stability. Recently, Liu et al. used trioctylphosphine as a ligand to achieve a stable and highly reactive PbI_2_ precursor, and further obtained CsPbI_3_ PQDs with a superior photoluminescence quantum yield of up to 100% [20]. Moreover, doping with additives is a generally efficient approach that not only enhances the charge transport properties but also controls the size and uniformity. In particular, creating an iodine-rich chemical environment during the PQD nucleation/growth process is favorable for minimizing the defect density. In addition, the existence of metal halides in the Pb-precursor could avoid the requirement of excess PbI_2_ for achieving high-quality PQDs [21]. Herein, we develop an easy synthetic method for colloidal FAPbI_3_ PQDs under atmospheric pressure in air with the assistance of ZnI_2_ additives. The air-synthesized Zn:FAPbI_3_ PQDs exhibit high structural and colloidal stability. By increasing the amount of ZnI_2_ additives, the crystal size of the FAPbI_3_ PQDs is decreased, while no significant changes in the crystal shape and size dispersion are observed. Surprisingly, with the addition of an optimal amount of ZnI_2_, ~30 mol%, the yielded FAPbI_3_ PQDs possess a longer carrier lifetime than that obtained by the vacuum/N_2_-assisted method. This synthetic method described here provides an efficient, simple, and robust approach to the low-cost production of high-performance FAPbI_3_ PQDs.

## 2. Materials and Methods

Oleylamine (OAm; technical grade, 70%) and oleic acid (OA; technical grade, 90%) were purchased from Sigma-Aldrich, Co., St. Louis, MO, USA. 1-octadecene (1-ODE; technical grade, 90%) was bought from Acros Organics, Doral, NW, USA. Hexane (GC, ≥96%) and formamidinium acetate (FA-acetate, 99%) were purchased from Tokyo chemical industry, Fukaya, Japan. PbI_2_ (99.9%) and ZnI_2_ (99.99%) were purchased from Advanced Election Technology Co., Ltd., Yingkou, Liaoning, China and Aladdin Biochemical Technology Co., Ltd., Shanghai, China, respectively. Methyl acetate (MeOAc, Super Dry, 99%) was purchased from J&K Scientific., Beijing, China. All of them were used directly without further purification.

X-ray diffraction (XRD) patterns were obtained with Rigaku Ultima IV (Tokyo, Japan, Cu Kα radiation, λ = 1.5418 Å) at room temperature in the air. For XRD measurements, the FAPbI_3_ PQD solution was concentrated by flowing nitrogen and dropped on the sample holder. Transmission electron microscope (TEM) was recorded with JEM-2800 (JEOL, Japan). UV-visible absorption spectra (UV-Vis) were measured with a Cary 100 spectrometer (Varian, Palo Alto, CA, USA) in absorbance mode. Steady-state photoluminescence (PL) emissions and time-resolved photoluminescence (TRPL) were carried out by Edinburgh FS5 (Edinburgh Instruments Ltd., Livingston, UK) with a monochromatized Xe lamp as the source excitation and the QD samples were excited at 475 nm.

## 3. Results and Discussion

FAPbI_3_ PQDs are commonly synthesized via a hot-injection method, as illustrated in Figure 1a. In detail, the FA-oleate precursor is prepared by mixing 0.521 g FAAc (5 mmol) with 10 mL OA and then the mixture is degassed under vacuum at 70 °C for 30 min. The temperature is subsequently increased to 110 °C and held to obtain a clear solution. Then, the FA-oleate precursor is maintained in nitrogen at 90 °C. Meanwhile, a mixture of 0.344 g PbI_2_ (0.75 mmol) and 20 mL 1-ODE is degassed under vacuum at 120 °C for 30 min in a three-necked round-bottom flask. Then, 4 mL OA and 2 mL OAm are heated to 120 °C for 20 min and added into the mixture of PbI_2_ and ODE. After PbI_2_ is fully dissolved, the temperature of the mixture is reduced to 80 °C under N_2_. For air-synthesized PQDs, the FA-oleate precursor and the Pb source are prepared by directly increasing the temperature to 80 °C in ambient air. Lastly, a 5 mL FA-oleate precursor solution is swiftly injected into the PbI_2_ mixture at 80 °C. After around 15 s, the reaction is quenched and cooled to room temperature using an ice water bath. For the purification of as-synthesized FAPbI_3_ PQDs, 9 mL MeOAc is added to the reaction mixture, which is subsequently centrifuged at 8000 rpm for 30 min. The precipitate is dispersed in 9 mL hexane, re-precipitated with 10 mL MeOAc, and centrifuged at 8000 rpm for 10 min. The final precipitate is dispersed in 2 mL octane and stored at room temperature.

The FAPbI_3_ PQDs synthesized under vacuum and N_2_ are denoted by FAPbI_3_-vaccum/N_2_. The vacuum-free air-synthesized FAPbI_3_ PQDs obtained without the addition of ZnI_2_ are denoted by Zn:FAPbI_3_-0. When different amounts of ZnI_2_ are added into the PbI_2_ precursor with the ZnI_2_/PbI_2_ molar percentage of 10% (0.075 mmol ZnI_2_), 20% (0.15 mmol ZnI_2_), 30% (0.225 mmol ZnI_2_), 40% (0.3 mmol ZnI_2_), and 50% (0.375 mmol ZnI_2_), the air-synthesized FAPbI_3_ PQDs prepared by the same synthetic method (Figure 1b) are denoted by Zn:FAPbI_3_-10, Zn:FAPbI_3_-20, Zn:FAPbI_3_-30, Zn:FAPbI_3_-40, and Zn:FAPbI_3_-50, respectively. After 20 days’ storage, the Zn:FAPbI_3_ PQDs show high colloidal stability, as demonstrated in Figure 1c. Furthermore, compared with the FAPbI_3_-vaccum/N_2_ PQDs, the Zn:FAPbI_3_ PQDs demonstrate brighter emission under UV illumination (395 nm), as observed in Figure 1c, which suggests that the introduction of ZnI_2_ may lead to reduced non-radiative recombination. The higher colloidal stability of the air-synthesized FAPbI_3_ PQDs can be attributed to the iodine-rich negatively charged surfaces and the decreased crystal size induced by the addition of ZnI_2_, as discussed below. To compare the production yield of PQDs, we add 5 μL of PQD solutions under different synthetic conditions into 3 mL hexane and measure their UV-Vis absorbance, as presented in Figure 1d,e. The absorbance of PQDs is increased by increasing the molar percentage of ZnI_2_/PbI_2_ from 0% to 30%. When the molar percentage is further increased to 40% and 50%, a decrease in absorbance can be observed. To clearly demonstrate this trend, we plot the absorbance at 500 nm, 600 nm, and 700 nm for different molar ratios of ZnI_2_/PbI_2_ in Figure 1e. It can be found that the Zn:FAPbI_3_-30 PQDs exhibit a higher production yield than other conditions, which results from their higher colloidal stability. However, the excessive addition of ZnI_2_ might suppress the formation of [PbI_6_]^4−^ octahedra by reducing the relative concentration of Pb^2+^ ions during QD nucleation and growth. The improved production yield and stability are helpful in reducing the cost of PQDs and the susceptibility to ambient environment which can be seen as the main obstacles for the further industrialization of PQDs, respectively.

First, we investigate the effect of ZnI_2_ additives on the morphologies of the FAPbI_3_ PQDs. The TEM measurements are performed and used to characterize the mean size and size distribution of the PQDs. As presented in Figure 2, the mean size is decreased by increasing the amount of added ZnI_2_, while there are no observable changes in the crystal shape and size dispersion of the FAPbI_3_ PQDs. Similarly, it has been reported for the CsPbX_3_ QDs that the particle size is decreased with increasing the X-to-Pb ratio in the reaction mixture, where the excess halide ions, X^−^, are able to suppress the further growth of QDs [21,22]. Therefore, the introduced ZnI_2_ can be seen as an excess source of I^−^, leading to the reduced PQD size. Nevertheless, it should be noted that when the added Zn salts have different halide ions than the PQDs, the over-all size changes may occur in an opposite trend. For example, during the synthesis of the CsPbI_3_ PQDs, the increase of the ZnCl_2_ additive amount results in a noticeable increase in crystal size, exhibiting a different influence of ZnI_2_ additives on the QD nucleation/growth mechanism [23]. Interestingly, we also observed that the ZnI_2_-free air-synthesized FAPbI_3_ PQDs (Zn:FAPbI_3_-0) demonstrate a smaller crystal size than the vacuum/N_2_-synthesized FAPbI_3_ PQDs. This reduction of QD size might be attributed to the existence of the water in the reaction solution as an antisolvent that is not removed in the air-synthesis process, thus facilitating the supersaturation of the PQDs.

The influence of the ZnI_2_ additive on the crystal structure of the FAPbI_3_ PQD is identified by the XRD measurements. In Figure 3, the vacuum/N_2_-synthesized FAPbI_3_ PQDs show the diffraction peaks at 13.7°, 27.9°, 31.1°, 39.9°, and 42.4° that can be assigned to (001), (002), (012), (022), and (003) plane of cubic α-phase perovskite, respectively [17]. The air-synthesized Zn:FAPbI_3_ PQDs do not show any new diffraction peaks (the left panel in Figure 3), while no significant peak shift is observed when the ZnI_2_ additive is introduced (the right panel in Figure 3). Since the ionic radius of Zn^2+^ (74 pm) is smaller than that of Pb^2+^ (119 pm), the replacement of Pb^2+^ by Zn^2+^ will bring a reduced lattice constant and thus higher-angle diffraction peaks [22,24,25]. However, if the added Zn^2+^ resides in an interstitial lattice position, this will lead to a lattice expansion, demonstrating a lower-angle diffraction peaks [26]. Both circumstances have been reported and the differences are probably attributed to the introduced ions or the types of A-site cations. It will definitely be interesting to identify the dynamic and thermodynamic processes during the Zn salt-involved PQD synthesis. However, based on the XRD results here, we state that the introduced ZnI_2_ would not induce the significant lattice distortion in the PQDs, rather only reside at the surface.

Lastly, we compare the optical and optoelectronic properties of these synthesized PQDs using different methods. The variations of PL emission intensity are presented in Figure 4a: Zn:FAPbI_3_-20 > Zn:FAPbI_3_-10 ≈ Zn:FAPbI_3_-0 > Zn:FAPbI_3_-30 > FAPbI_3_-vaccum/N_2_ > Zn:FAPbI_3_-40 > Zn:FAPbI_3_-50 PQDs. The PL emission intensity of the colloidal QDs is influenced by many factors: solution concentration, particle size, particle dispersity, ligand binding, and defect density. As observed in Figure 4a, the variation in PL intensity for different FAPbI_3_ PQDs is complex and is a result of competitions among these factors. Thus, we are only able to presume that the better colloidal stability and decreased iodine-vacancy states are helpful in enhancing the PL intensity, while the weakened ligand binding and formed impurities induced by excessive addition of ZnI_2_ result in a decrease in PL intensity. Therefore, the higher intensity of Zn:FAPbI_3_-0, Zn:FAPbI_3_-10, Zn:FAPbI_3_-20, and Zn:FAPbI_3_-30 than the pristine FAPbI_3_-vaccum/N_2_ PQDs may be due to suppressed non-radiative recombination and improved particle dispersity. However, when excessive ZnI_2_ is used, the Zn ions start to negatively impact the ligand binding with OA and OAm, inducing the defects on the PQD surface, further resulting in a decrease in PL intensity. In addition to intensity variation, the peak position shifts are also observed in Figure 4b. The blue shift can be attributed to the strong quantum confinement that is demonstrated by the FAPbI_3_ PQDs with a smaller crystal size (as illustrated in Figure 2). The smaller size of PQDs typically results in stronger quantum confinement effects, showing a higher bandgap (a higher-energy PL emission) [27]. Zn:FAPbI_3_-30, Zn:FAPbI_3_-40, and Zn:FAPbI_3_-50 PQDs exhibit red shifts in a peak position (a lower-energy PL emission), which is likely caused by the weakened electronic coupling between QDs and the formation of impurities of related Zn salts on PQD surfaces owing to the excessive Zn^2+^ [28]. Furthermore, TRPL measurements are carried out to examine the carrier lifetime, as demonstrated in Figure 4c. Two TRPL lifetimes, *τ*_1_ and *τ*_2_, are plotted with different synthesis conditions of FAPbI_3_ PQDs in Figure 4d, which are drawn from the fitting results using a two-exponential decay function. The fast decay component *τ*_1_ and the slow decay component *τ*_2_ are ascribed to a trap-assisted recombination at the QD surfaces and an intrinsic radiative recombination inside the QDs, respectively [4,29]. It should be noted that the average lifetime is not presented here, because the average lifetime basically has no real physical meaning. Both *τ*_1_ and *τ*_2_ for the air-synthesized FAPbI_3_ PQDs are decreased, compared with the PQDs obtained using degassing (vacuum) and N_2_ protection, which can be ascribed to the moisture and volatile contaminants in the reaction mixture inducing large amounts of defects in the PQDs. However, the lifetime of the photogenerated charge carrier, both *τ*_1_ and *τ*_2_, can be prolonged by introducing the ZnI_2_ additive, as illustrated in Figure 4d. The Zn:FAPbI_3_-30 PQDs exhibit 29.81 ns for *τ*_1_ and 96.04 ns for *τ*_2_, comparable with, or slightly higher compared with, the pristine FAPbI_3_-vaccum/N_2_ PQDs (28.05 ns for *τ*_1_ and 92.54 ns for *τ*_2_). The iodine-rich environment provided by the addition of ZnI_2_ inhibits the formation of iodine-vacancy trap states in the FAPbI_3_ PQDs, which is also beneficial to the structural stability [4]. Nevertheless, when the percentage of ZnI_2_ is increased to 40% and 50%, the increased surface-to-volume ratio (due to the decrease in size) and the abundant Zn ions that can be seen as new contaminants, lead to a new defect state formation inside and on the PQD surfaces, which is responsible for the decreased TRPL lifetimes shown in Figure 4d. For practical device applications, the enhanced carrier lifetime of PQDs may lead to increased current density in solar cells, increased external quantum efficiency in LEDs, and improved responsivity in photodetectors, etc.

## 4. Conclusions

In summary, an easy and effective synthetic method by introducing ZnI_2_ additives is developed to achieve high-quality FAPbI_3_ PQDs without degassing and N_2_ protection. Compared with the pristine vacuum/N_2_-FAPbI_3_ PQDs, the air-synthesized Zn:FAPbI_3_ PQDs exhibit higher structural and colloidal stability, as well as higher production yield. As the amount of added ZnI_2_ in the reaction mixture is increased, the crystal size of the FAPbI_3_ PQDs is decreased, while no significant changes in the crystal structure, shape, and size dispersion are observed. The introduced ZnI_2_ would not induce the significant lattice distortion in the PQDs, but may only reside at the surfaces. Thus, the excessive addition of ZnI_2_ induces the formation of impurities of related Zn salts on PQD surfaces, negatively impacts the ligand binding with OA and OAm, and results in defects as well as weakened electronic coupling between QDs. However, importantly, with assistance of an optimal amount of ZnI_2_ additives, ~30 mol%, these air-synthesized FAPbI_3_ PQDs demonstrate slightly longer carrier lifetimes than vacuum/N_2_-assisted FAPbI_3_ PQDs. This vacuum-free air-synthesis route offers an industrial-friendly method for producing low-cost colloidal FAPbI_3_ PQDs for high-performance PV applications.

## Figures and Tables

**Figure 1 nanomaterials-13-00226-f001:**
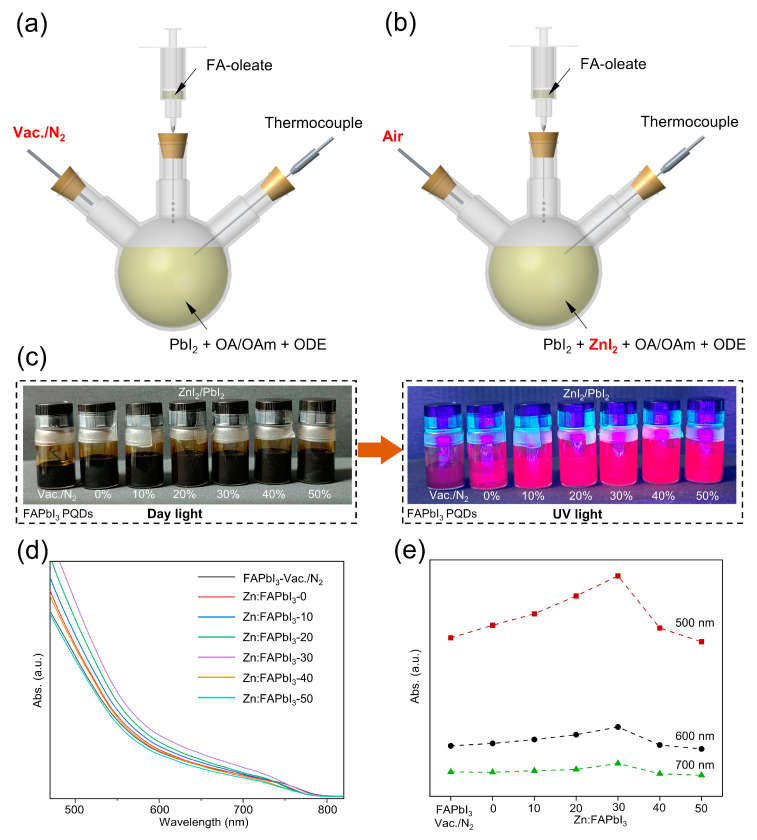
Scheme of the hot injection of FAPbI_3_ PQDs under N_2_ (**a**) and in air (**b**). Photographs of FAPbI_3_ PQDs solutions with same concentrations but different synthesis conditions under daylight and UV light (**c**). The UV-Vis spectra (**d**) and the absorbance at 500 nm, 600 nm, and 700 nm (**e**) for FAPbI_3_-vaccum/N_2_, Zn:FAPbI_3_-0, Zn:FAPbI_3_-10, Zn:FAPbI_3_-20, Zn:FAPbI_3_-30, Zn:FAPbI_3_-40, and Zn:FAPbI_3_-50 PQDs.

**Figure 2 nanomaterials-13-00226-f002:**
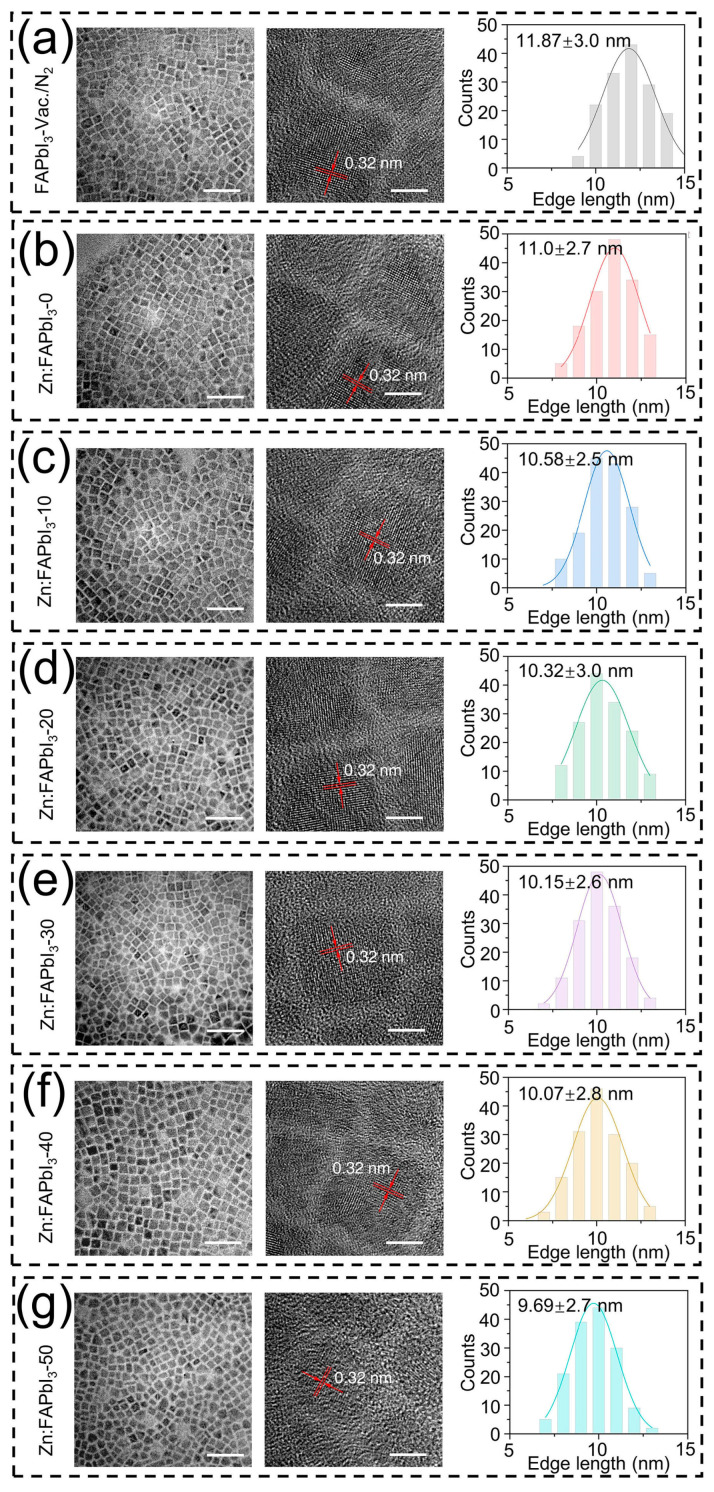
TEM images and size distributions with Gauss fitting of FAPbI_3_-vaccum/N_2_ (**a**), Zn:FAPbI_3_-0 (**b**), Zn:FAPbI_3_-10 (**c**), Zn:FAPbI_3_-20 (**d**), Zn:FAPbI_3_-30 (**e**), Zn:FAPbI_3_-40 (**f**), and Zn:FAPbI_3_-50 (**g**) PQDs. The scale bars in the left and right TEM images are 50 nm and 5 nm, respectively. The size distributions are obtained by counting 150 particles.

**Figure 3 nanomaterials-13-00226-f003:**
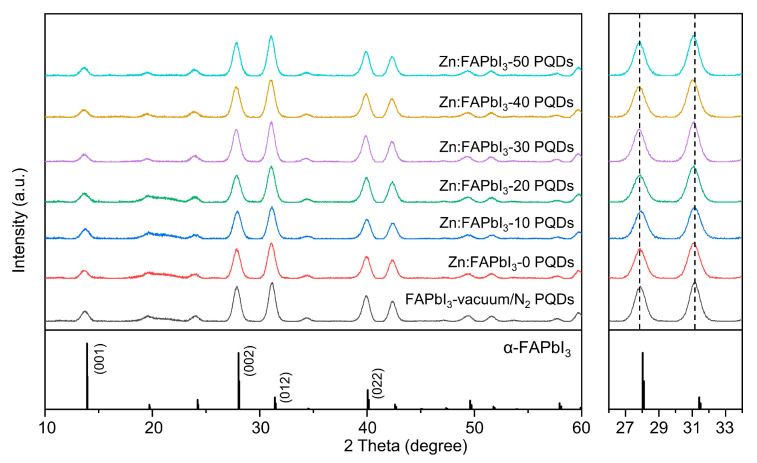
XRD patterns of the FAPbI_3_-vaccum/N_2_, Zn:FAPbI_3_-0, Zn:FAPbI_3_-10, Zn:FAPbI_3_-20, Zn:FAPbI_3_-30, Zn:FAPbI_3_-40, and Zn:FAPbI_3_-50 PQDs. The right panel presents the enlarged XRD patterns in the range of 26 to 34 degrees.

**Figure 4 nanomaterials-13-00226-f004:**
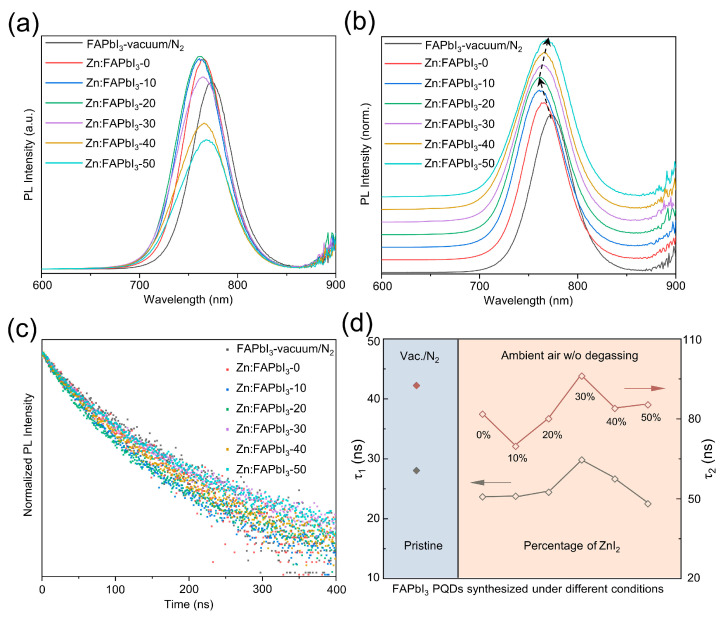
PL (**a**), normalized PL (**b**) spectra, and TRPL data (**c**) for the FAPbI_3_-vaccum/N_2_, Zn:FAPbI_3_-0, Zn:FAPbI_3_-10, Zn:FAPbI_3_-20, Zn:FAPbI_3_-30, Zn:FAPbI_3_-40, and Zn:FAPbI_3_-50 PQDs. Summary of TRPL decay times (**d**) for FAPbI_3_ PQDs obtained under different synthetic conditions.

## Data Availability

Data that support the plots within this work are available from the corresponding author upon reasonable request.

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
