# Peer review of "A Low-Cost Synthetic Route of FAPbI3 Quantum Dots in Air at Atmospheric Pressure: The Role of Zinc Iodide Additives"

_nanomaterials, 2023, doi:10.3390/nano13020226_

Round 1
Reviewer 1 Report
This paper describes low-cost synthetic method to synthesize FAPbI3 Perovskite quantum dots (PQDs) in air at atmospheric pressure with the assistance of ZnI2. Some questions are the following:
1) Throughout the paper it is not explained why the air-synthesized Zn:FAPbI3 PQDs exhibit higher structural and colloidal stability than pristine vacuum/N2-FAPbI3 PQDs which is one of the main conclusions of the paper.
2) The finding that Zn:FAPbI3-30 PQDs exhibit higher production yield than other conditions has not been satisfactorily motivated. This is also another main conclusion of the paper.
3) What are the conclusions in relation to the influence of ZnI2 during the synthesis process on morphologies and optoelectronic properties?
4) The variations of PL emission intensity are presented in Figure 4a should be better explained. Why is Zn:FAPbI3-20 > Zn:FAPbI3-10 ≈ Zn:FAPbI3-0 > Zn:FAPbI3-30 > FAPbI3-vac-cum/N2 > Zn:FAPbI3-40 > Zn:FAPbI3-50 PQDs?
4a) The first comment provided by the authors is about the higher intensity of Zn:FAPbI3-0, Zn:FAPbI3-10, Zn:FAPbI3-20, and Zn:FAPbI3-30 than pristine FAPbI3-vaccum/N2 PQDs that may be due to suppressed non-radiative recombination and improved particle dispersity. Why is Zn:FAPbI3-20 > Zn:FAPbI3-10 ≈ Zn:FAPbI3-0 > Zn:FAPbI3-30 ?
4b) The second comment provided by the authors is that when excessive ZnI2 is used, the Zn ions start to negatively impact the ligand binding with OA and OAm, inducing the defects on PQD surface, further resulting in a decrease in PL intensity. How could this hypotesis be tested? Why is FAPbI3-vac-cum/N2 > Zn:FAPbI3-40 > Zn:FAPbI3-50 PQDs ?
5) The peak position shifts in Figure 4b are not satisfactorily explained. References are cited but the explanatory text provided is very short.
6) It is not explained why once the ZnI2 additive is introduced, the lifetime of photogenerated charge carrier is prolonged. Are there any practical implications for such a difference?
7) The practical implication of the air-synthesized Zn:FAPbI3 PQDs 216 exhibiting higher structural and colloidal stability, as well as higher production yield are not explained.
8) The conclusions are rather short and do not satisfactorily summarize all the findings of the paper.
Author Response
This paper describes low-cost synthetic method to synthesize FAPbI3 Perovskite quantum dots (PQDs) in air at atmospheric pressure with the assistance of ZnI2. Some questions are the following:
1) Throughout the paper it is not explained why the air-synthesized Zn:FAPbI3 PQDs exhibit higher structural and colloidal stability than pristine vacuum/N2-FAPbI3 PQDs which is one of the main conclusions of the paper.
Author reply: Thanks for the suggestion. The explanations have been presented in the revised manuscript, as “The higher colloidal stability of air-synthesized FAPbI3 PQDs can be attributed to the iodine-rich negatively charged surfaces and the decreased crystal size induced by the addition of ZnI2, as discussed below.” “The iodine-rich environment provided by the addition of ZnI2 inhibits the formation of iodine-vacancy trap states in FAPbI3 PQDs, which is also beneficial to the structural stability.”
2) The finding that Zn:FAPbI3-30 PQDs exhibit higher production yield than other conditions has not been satisfactorily motivated. This is also another main conclusion of the paper.
Author reply: Thanks for the suggestion. For the synthesis of colloidal quantum dot, the higher production yield is usually attributed to higher colloidal stability. In other words, the agglomeration of PQDs is suppressed by iodine-rich negatively charged surfaces and reduced QD size, yielding colloidally stable PQDs. The detailed description has been added in the revised version, “It can be found that Zn:FAPbI3-30 PQDs exhibit higher production yield than other conditions, which is resulted from their higher colloidal stability. However, the excessive addition of ZnI2 might suppress the formation of [PbI6]4– octahedra by reducing the relative concentration of Pb2+ ions during QD nucleation and growth.”
3) What are the conclusions in relation to the influence of ZnI2 during the synthesis process on morphologies and optoelectronic properties?
Author reply: For morphologies, the mean size is decreased as increasing the amount of added ZnI2, while there are no observable changes in the crystal shape and size dispersion of FAPbI3 PQDs. The decreased particle size is due to the increased X-to-Pb ratio in the reaction mixture by the addition of ZnI2, where the excess halide ions, X-, are able to suppress the further growth of QDs. In regard to optoelectronic properties, the main conclusion is that the Zn:FAPbI3-30 PQDs exhibit comparable or slightly higher carrier lifetime compared to pristine FAPbI3-vaccum/N2 PQDs, which is ascribed to the suppressed formation of iodine-vacancy trap states in PQDs.
The conclusion is modified as “In summary, an easy and effective synthetic method by introducing ZnI2 additives is developed to achieve high-quality FAPbI3 PQDs without degassing and N2 protection. Compared to the pristine vacuum/N2-FAPbI3 PQDs, the air-synthesized Zn:FAPbI3 PQDs exhibit higher structural and colloidal stability, as well as higher production yield. As the amount of added ZnI2 in reaction mixture is increased, the crystal size of FAPbI3 PQDs is decreased, while no significant changes in the crystal structure, shape and size dispersion are observed. The introduced ZnI2 would not induce the significant lattice distortion in PQDs, but may only reside at the surfaces. Thus, the excessive addition of ZnI2 induces the formation of impurities of related Zn salts on PQD surfaces, negatively impacts the ligand binding with OA and OAm, and results in defects as well as weakened electronic coupling between QDs. But, importantly, with assistance of an optimal amount of ZnI2 additives, ~30 mol%, these air-synthesized FAPbI3 PQDs demonstrate slightly longer carrier lifetime than vacuum/N2-assisted FAPbI3 PQDs. This vacuum-free air-synthesis route offers an industrial-friendly method to produce low-cost colloidal FAPbI3 PQDs for high-performance PV applications.”
4) The variations of PL emission intensity are presented in Figure 4a should be better explained. Why is Zn:FAPbI3-20 > Zn:FAPbI3-10 ≈ Zn:FAPbI3-0 > Zn:FAPbI3-30 > FAPbI3-vac-cum/N2 > Zn:FAPbI3-40 > Zn:FAPbI3-50 PQDs?
4a) The first comment provided by the authors is about the higher intensity of Zn:FAPbI3-0, Zn:FAPbI3-10, Zn:FAPbI3-20, and Zn:FAPbI3-30 than pristine FAPbI3-vaccum/N2 PQDs that may be due to suppressed non-radiative recombination and improved particle dispersity. Why is Zn:FAPbI3-20 > Zn:FAPbI3-10 ≈ Zn:FAPbI3-0 > Zn:FAPbI3-30 ?
4b) The second comment provided by the authors is that when excessive ZnI2 is used, the Zn ions start to negatively impact the ligand binding with OA and OAm, inducing the defects on PQD surface, further resulting in a decrease in PL intensity. How could this hypotesis be tested? Why is FAPbI3-vac-cum/N2 > Zn:FAPbI3-40 > Zn:FAPbI3-50 PQDs ?
Author reply: Sorry for the misunderstanding and non-clear descriptions on the variations in PL emission intensity, since the PL emission intensity of colloidal QDs is influenced by many factors: the solution concentration, particle size, colloidal stability, ligand binding, and defect density. The variation in PL intensity for different FAPbI3 PQDs is complex. Thus, we are only able to presume why, without proceeding much more precise experiments and advanced characterizations. We believe that the better colloidal stability and decreased iodine-vacancy states are helpful to enhance the PL intensity, while the weakened ligand binding and the formed impurity induced by excessive addition of ZnI2 result in the decrease of PL intensity. Therefore, from FAPbI3-vaccum/N2, Zn:FAPbI3-0, Zn:FAPbI3-10 to Zn:FAPbI3-20 PQDs, the intensity is increased, which is probably attributed to improved colloidal stability, decreased particle size or/and decreased iodine-vacancy states. Nevertheless, from Zn:FAPbI3-30, Zn:FAPbI3-40 to Zn:FAPbI3-50 PQDs, the intensity is decreased, which is likely resulted from the weakened ligand binding and the formed impurity induced by excessive addition of ZnI2. Overall, the relative variation of PL intensity is a result of competition among these factors. Accordingly, the explanations are modified as “The PL emission intensity of colloidal QDs is influenced by many factors: solution concentration, particle size, particle dispersity, ligand binding, and defect density. As observed in Figure 4a, the variation in PL intensity for different FAPbI3 PQDs is complex, and it is a result of competitions among these factors. Thus, we are only able to presume that the better colloidal stability and decreased iodine-vacancy states are helpful to enhance the PL intensity, while the weakened ligand binding and formed impurities induced by excessive addition of ZnI2 result in a decrease of PL intensity.”
5) The peak position shifts in Figure 4b are not satisfactorily explained. References are cited but the explanatory text provided is very short.
Author reply: Thanks for the comment. The peak position shifts are further explained in details, as “The blue shift can be attributed to the strong quantum confinement that is demonstrated by FAPbI3 PQDs with smaller crystal size (as illustrated in Figure 2). The smaller size of PQDs typically results in stronger quantum confinement effects, showing a higher bandgap (a higher-energy PL emission). [26] Zn:FAPbI3-30, Zn:FAPbI3-40 and Zn:FAPbI3-50 PQDs exhibit red shifts in peak position (a lower-energy PL emission), which is likely caused by the weakened electronic coupling between QDs and the formation of impurities of related Zn salts on PQD surfaces owing to the excessive Zn2+.[27]”
6) It is not explained why once the ZnI2 additive is introduced, the lifetime of photogenerated charge carrier is prolonged. Are there any practical implications for such a difference?
Author reply: The iodine-rich environment provided by the addition of ZnI2 inhibits the formation of iodine-vacancy trap states in FAPbI3 PQDs. As the trap density in PQDs is reduced, the trap-assisted non-radiative recombination is suppressed, which further leads to the prolonged TRPL lifetime. The long carrier lifetime is favorable for high-performance photovoltaic devices, LEDs, lasers, etc. For example, the improved carrier lifetime can result in longer carrier diffusion length, more efficient photogenerated charge collection, and higher PL quantum yield, further leading to higher current density in solar cells, higher external quantum efficiency in LEDs, etc. The modifications are added in the revised paper, as “For practical device application, the enhanced carrier lifetime may lead to increased current density in solar cells, increased external quantum efficiency in LEDs, and improved responsivity in photodetectors, etc.”
7) The practical implication of the air-synthesized Zn:FAPbI3 PQDs 216 exhibiting higher structural and colloidal stability, as well as higher production yield are not explained.
Author reply: Thank you for your suggestion on practical implication. PQDs exhibit outstanding optoelectronic properties and can be widely used in solar cells, light-emitting diodes, photodetectors and lasers. However, PQDs are highly susceptible to polar solvent, light, thermal and so on, which is detrimental to their practical applications. Therefore, the improvement of the stability is urgent and meaningful. On the other hand, the cost of PQDs can be decreased by enhancing production yield. The modifications are added in the revised paper, as “The improved production yield and stability are helpful to reduce the cost of PQDs and the susceptibility to ambient environment which can be seen as main obstacles for further industrialization of PQDs, respectively.”
8) The conclusions are rather short and do not satisfactorily summarize all the findings of the paper.
Author reply: The conclusions have been revised as “In summary, an easy and effective synthetic method by introducing ZnI2 additives is developed to achieve high-quality FAPbI3 PQDs without degassing and N2 protection. Compared to the pristine vacuum/N2-FAPbI3 PQDs, the air-synthesized Zn:FAPbI3 PQDs exhibit higher structural and colloidal stability, as well as higher production yield. As the amount of added ZnI2 in reaction mixture is increased, the crystal size of FAPbI3 PQDs is decreased, while no significant changes in the crystal structure, shape and size dispersion are observed. The introduced ZnI2 would not induce the significant lattice distortion in PQDs, but may only reside at the surfaces. Thus, the excessive addition of ZnI2 induces the formation of impurities of related Zn salts on PQD surfaces, negatively impacts the ligand binding with OA and OAm, and results in defects as well as weakened electronic coupling between QDs. But, importantly, with assistance of an optimal amount of ZnI2 additives, ~30 mol%, these air-synthesized FAPbI3 PQDs demonstrate slightly longer carrier lifetime than vacuum/N2-assisted FAPbI3 PQDs. This vacuum-free air-synthesis route offers an industrial-friendly method to produce low-cost colloidal FAPbI3 PQDs for high-performance PV applications.”
Reviewer 2 Report
For high-efficient industrial solar cells, affordable and secure growing methods are necessary. The authors proposed a method for the FAPbI3 perovskite quantum dots synthesis in air and introducing ZnI2. The presented article is very well-written and clear.
Some minor notes:
-- page 67 Instead of "facile", its synonym "easy" can be used.
-- Reference suggestion: DOI: 10.1002/solr.202200120.
Author Response
For high-efficient industrial solar cells, affordable and secure growing methods are necessary. The authors proposed a method for the FAPbI3 perovskite quantum dots synthesis in air and introducing ZnI2. The presented article is very well-written and clear.
Some minor notes:
-- page 67 Instead of "facile", its synonym "easy" can be used.
-- Reference suggestion: DOI: 10.1002/solr.202200120.
Author reply: Thanks for your kind comments. In Line 67, the word has been replaced by “easy”. Also, the mentioned reference has been added as Ref. 10 in the revised manuscript.
Reviewer 3 Report
This work by Wang et al. is a well laid out study with the approach not entirely novel but relevant to assess the photoactivity of PQDs for real world applications, synthesized under air. This aspect makes this study suitable for publication in this journal as scientifically this study is also sound in addition to being applicable and is indeed well written. Seemingly marginal improvements are attained for the air prepared PQDs vs. controlled atmosphere but that is important since matching the performance between the two conditions is not trivial. As such, this reviewer recommends publication with minimal revision listed below that can be of use also for authors’ future work.
1. Synthetic procedure is well presented but authors may want to mention the moles of reagents used as it will be valuable to a boarder audience in the field.
2. Please avoid using phrases such as ‘As we all know…’, kindly replace it simply saying ‘Since the ionic radius of…’
3. Please read and grammatically correct sentences in lines 200-204 on page 7 of the submitted version.
4. TEM images are consistent and measurements statistically and the size decreases as ZnI2 dopant is increased? Have authors been able to identify d-spacings for doped FAPbI3 and associated trends? The XRD data is very similar between samples which makes the possibility of spacing measurements somewhat tricky. Also, it is a reasonable observation that under air sizes are smaller since water is indeed an etchant/antisolvent.
Author Response
Reviewer 2
This work by Wang et al. is a well laid out study with the approach not entirely novel but relevant to assess the photoactivity of PQDs for real world applications, synthesized under air. This aspect makes this study suitable for publication in this journal as scientifically this study is also sound in addition to being applicable and is indeed well written. Seemingly marginal improvements are attained for the air prepared PQDs vs. controlled atmosphere but that is important since matching the performance between the two conditions is not trivial. As such, this reviewer recommends publication with minimal revision listed below that can be of use also for authors’ future work.
- Synthetic procedure is well presented but authors may want to mention the moles of reagents used as it will be valuable to a boarder audience in the field.
Author reply: Thank you for your kind suggestion. The moles of reagents are added as “In detail, FA-oleate precursor is prepared by mixing 0.521 g FAAc (5 mmol) with 10 mL OA and then the mixture is… Meanwhile, a mixture of 0.344 g PbI2 (0.75 mmol) and 20 mL 1-ODE is degassed …When different amounts of ZnI2 are added into PbI2 precursor with the ZnI2/PbI2 molar percentage of 10% (0.075 mmol ZnI2), 20% (0.15 mmol ZnI2), 30% (0.225 mmol ZnI2), 40% (0.3 mmol ZnI2) and 50% (0.375 mmol ZnI2), the air-synthesized FAPbI3 PQDs, prepared by the same synthetic method (Figure 1b), are denoted by Zn:FAPbI3-10, Zn:FAPbI3-20, Zn:FAPbI3-30, Zn:FAPbI3-40, and Zn:FAPbI3-50, respectively.”
- Please avoid using phrases such as ‘As we all know…’, kindly replace it simply saying ‘Since the ionic radius of…’
Author reply: Thanks for the suggestion. The sentence is modified as “Since the ionic radius of Zn2+ (74 pm) is smaller than that of Pb2+ (119 pm), and therefore the replacement of Pb2+ by Zn2+ will bring a reduced lattice constant and thus to higher-angle diffraction peaks.”
- Please read and grammatically correct sentences in lines 200-204 on page 7 of the submitted version.
Author reply: The sentences are changed to “It should be noted that the average lifetime is not presented here, and that is because the average lifetime basically has no real physical meaning. Both τ1 and τ2 for the air-synthesized FAPbI3 PQDs are decreased compared to the PQDs obtained using degassing (vacuum) and N2 protection, which can be ascribed to the moisture and volatile contaminants in reaction mixture inducing large amounts of defects in PQDs. However, the lifetime of photogenerated charge carrier, both τ1 and τ2, can be prolonged by introducing the ZnI2 additive, as illustrated in Figure 4d.”
- TEM images are consistent and measurements statistically and the size decreases as ZnI2 dopant is increased? Have authors been able to identify d-spacings for doped FAPbI3 and associated trends? The XRD data is very similar between samples which makes the possibility of spacing measurements somewhat tricky. Also, it is a reasonable observation that under air sizes are smaller since water is indeed an etchant/antisolvent.
Author reply: Thanks for the comments. As presented in size distribution of Figure 2, the mean size is decreased as increasing the amount of added ZnI2, while there are no observable changes in the crystal shape and size dispersion of FAPbI3 PQDs. In addition, we do not observe significant differences in XRD results between different FAPbI3 PQDs. We absolutely agree that extracting d-spacing information from these XRD data is tricky, may lead to misunderstanding. In another way, we further measured d-spacings for the FAPbI3 PQDs obtained under different synthetic conditions, and the results are shown in Figure 2. All FAPbI3 PQD samples demonstrate a d-spacing value of ~0.32nm.
Round 2
Reviewer 1 Report
The revised version of the paper satisfactorily answers the concerns raised to the first submission.